# Urban Vulnerability Assessment for Pandemic Surveillance—The COVID-19 Case in Bogotá, Colombia

**Jeisson Prieto** [1,*] **, Rafael Malagón** [2] **, Jonatan Gomez** [3] **and Elizabeth León** [3]

1   Departamento de Matemáticas, Facultad de Ciencias, Universidad Nacional de Colombia, Bogotá 11001, Colombia
2   Departamento de Salud Colectiva, Facultad de Odontología, Universidad Nacional de Colombia, Bogotá 11001, Colombia; ramalagono@unal.edu.co
3   Departamento de Ingeniería de Sistemas e Industrial, Facultad de Ingeniería, Universidad Nacional de Colombia, Bogotá 11001, Colombia; jgomezpe@unal.edu.co (J.G.); eleonguz@unal.edu.co (E.L.)
*   Correspondence: japrietov@unal.edu.co

**Abstract:** A pandemic devastates the lives of global citizens and causes significant economic, social, and political disruption. Evidence suggests that the likelihood of pandemics has increased over the past century because of increased global travel and integration, urbanization, and changes in land use with a profound affectation of society–nature metabolism. Further, evidence concerning the urban character of the pandemic has underlined the role of cities in disease transmission. An early assessment of the severity of infection and transmissibility can help quantify the pandemic potential and prioritize surveillance to control highly vulnerable urban areas in pandemics. In this paper, an Urban Vulnerability Assessment (UVA) methodology is proposed. UVA investigates various vulnerability factors related to pandemics to assess the vulnerability in urban areas. A vulnerability index is constructed by the aggregation of multiple vulnerability factors computed on each urban area (i.e., urban density, poverty index, informal labor, transmission routes). This methodology is useful in a-priori evaluation and development of policies and programs aimed at reducing disaster risk (DRR) at different scales (i.e., addressing urban vulnerability at national, regional, and provincial scales), under diverse scenarios of resources scarcity (i.e., short and long-term actions), and for different audiences (i.e., the general public, policy-makers, international organizations). The applicability of UVA is shown by the identification of high vulnerable areas based on publicly available data where surveillance should be prioritized in the COVID-19 pandemic in Bogotá, Colombia.

**Keywords:** Urban vulnerability; Vulnerability assessment; Infectious diseases; Pandemic; COVID-19 vulnerability index; Spatial analysis

## 1. Introduction

Pandemics are intercontinental-scale outbreaks of infectious diseases that increase morbidity and mortality over a big geographic area and cause significant social, political, and economical disruption [1,2]. Previous pandemics [1] have exposed gaps related to the timely detection of disease, tracing of contacts, availability of basic care, quarantine and isolation procedures, and health sector preparedness (i.e., global coordination and response mobilization) [3,4]. Suddenly, significant policy attention has focused on the need to identify and limit emerging outbreaks that might lead to pandemics and to expand and sustain investment to build preparedness and health capacity [5]. Nonetheless, the timeliness of implementing these measures is paramount to control a highly contagious disease. Efficient prioritization of investigation of highly vulnerable areas would help to optimize the use of resources and potentially limit the size of the pandemic [6–8].

Vulnerability Assessment describes the degree to which socioeconomic systems and physical assets in geographic areas are either susceptible or resilient to the impact of a disaster (i.e., pandemic). Once the vulnerability is evaluated across areas, it is possible to

prioritize them and undertake preventative action and response efforts (i.e., planning and coordination, reducing the spread of disease, continuity of health care provision) [1,2,9,10]. In the urban context, the Urban Vulnerability Assessment (UVA) helps to determine what types of preparedness and response activities might support an optimal Urban Strategic Planning (USP) to assist the decision-making processes [11].

Several models have been proposed to establish vulnerable urban areas over the infectious disease domain, that is, vector-borne diseases [12], Dengue [9], malaria [13,14], and Ebola [10]. Recently, in [15] a COVID-19 vulnerability index for urban areas in India was proposed which aggregates weighted scores of a set of variables related to COVID-19 precaution of social distance and lockdown in four metro cities in India. Nevertheless, relative preferences between criteria based judgments for the gathering of preferences for indicators (vulnerability factors) is needed in those models, having some limitations such as expert bias, or hierarchical criteria to weight the factors [16,17].

On the other hand, the recently UN-Habitat response plan for the current COVID-19 pandemic underlined the urban-centric character of the disease, indicating that above 95% of the cases are located in urban areas [18]. The World Health Organization (WHO) emphasized that the first transmission in the COVID-19 pandemic did happen in the internationally connected megacities [19]. Further, interconnected cities in South America (i.e., Bogotá) are presumably more susceptible given their population densities, low income, job informality, and lack of affordable health services [20].

In this paper, a conceptual framework for Urban Vulnerability Assessment (UVA) for pandemics is proposed. This UVA conducted a comprehensive review of relevant literature to identify vulnerability factors influencing pandemics. These were then condensed into an index that allowed us to establish and rank potentially vulnerable urban areas. The vulnerability rank is built using Borda's count aggregation method, which does not need experts knowledge nor additional parameters. UVA is framed in the current COVID-19 pandemic in Bogotá, the most densely populated city of Colombia. Using public available data of Bogotá, UVA creates a spatially explicit description of vulnerability for COVID-19 pandemic. This modeling application study provides a potential tool to inform policy-makers to prioritize resource allocation and devise effective mitigation and reconstruction strategies for affected populations in Bogotá.

This paper is divided into four sections. Section 2 develops the methodology of Urban Vulnerability Assessment (UVA) for pandemic surveillance. Section 3 describes the applicability of UVA for the current COVID-19 pandemic in Bogotá, Colombia. Finally, Section 4 discusses some of the conclusions and potential future developments.

## 2. Vulnerability Assessment

The conceptual framework of Urban Vulnerability Assessment (UVA) for pandemics surveillance is illustrated in Figure 1. UVA involves four main stages. The first stage is the identification of vulnerability factors influencing pandemics (Figure 1, panel (a)). The second stage is to transform the raw input data from each vulnerability factor into a probability distribution (Figure 1, panel (b)). The third stage groups geographic areas with similar characteristics into classes to assign a vulnerability level (Figure 1, panel (c)). After that, an aggregation method is applied to create a unique rank for each class (Figure 1, panel (d)), where a higher rank is assigned to a higher vulnerability level (Figure 1, panel (e)).

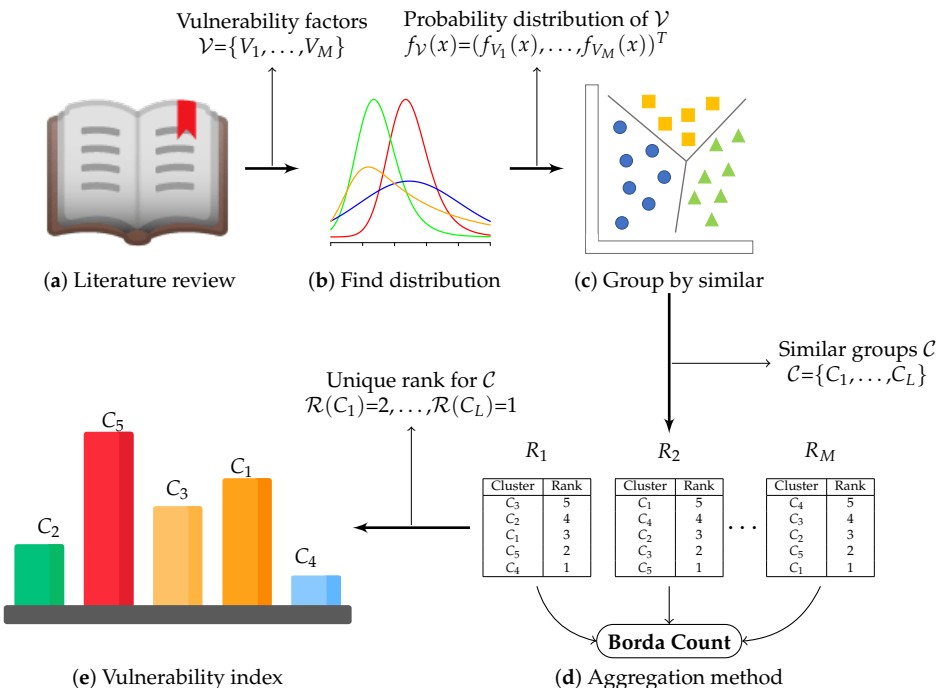

**Figure 1.** Schematic diagram of the Urban Vulnerability Assessment for Pandemic Surveillance.

## 2.1. Literature Review

We conducted a focused literature search [21] to identify a set of peer-reviewed studies that possibly examined types of vulnerability factors related to pandemics. The studies consider both factors related to past pandemics (i.e, 1881 Fifth cholera, 1918 Spanish flu influenza, 1957 Asian flu influenza, 2003 SARS, 2009 h1n1, 2013 West Africa Ebola) and factors found in the current COVID-19 pandemic. The literature search used a combination of search strings to retrieve studies in the Google Scholar database (i.e., (hazard OR uncertain* OR risk* OR vulnerab*) AND (disease OR pandemic* OR endemic*) AND (analysis OR factor* OR assess*)). The search included peer-reviewed english language journal articles (called "studies" in our review) published between 1982 and 2020. The retrieved studies for which the study's title, abstract, or keywords indicated the study examined a type of vulnerability in pandemics. Then, a manual assessment is made for every study against eligibility criteria:

- The study provided a quantitative or conceptual analysis of vulnerability factors related to infectious diseases (or pandemics).
- The core of the study included vulnerability.
- The study focuses on urban areas.
- The study focuses more on the vulnerability at the geographic area level than on the individual level.

Afterward, the vulnerability factor the study focused on, the geographic focus of the study and the methods used to assess the vulnerability were recorded. This involved examining the title, abstract, keywords, or full-text version. We also listed the country or region(s) where the study focused. For theoretical studies without a clear geographic focus, the geographic location is listed as Not Applicable (NA). Table 1 summarizes the 11 studies that were considered for the analysis of vulnerability factors related to pandemics.

**Table 1.** Summary of studies considered to vulnerability factors related with pandemics.

| Reference | Vulnerable Factor(s) | Geographic Focus | Methods | Main Findings |
|---|---|---|---|---|
| [22] | a. Essential worker<br>b. Household size<br>c. Age<br>d. Gender | Singapur | Demographic, clinical, treatment, and laboratory data | Describe incidence and vulnerability factors for pandemic in healthcare personnel |
| [1] | e. Geographic spark<br>f. Geographic spread<br>g. Burden quantification<br>h. Disease importation<br>a. Essential worker<br>i. Healthcare access | NA | Epidemiology evidence of previous infectious diseases | Covers the concerning of vulnerability, impacts, mitigation and pandemic knowledge gaps |
| [23] | j. Medical preconditions<br>c. Age<br>d. Gender<br>a. Essential worker | United Kingdom | Epidemiology evidence of COVID-19 | How the vulnerability might vary in different population groups or settings |
| [24] | k. Time delay illness<br>l. Insufficient follow-up<br>c. Age<br>d. Gender | China | Demographic, clinical, treatment, and laboratory data | Provides insights in early vulnerability assessment using publicly available data |
| [25] | i. Hospital capacity<br>m. Water and sanitation<br>n. Logistics<br>o. Per capita income<br>p. Public education | NA | Conceptual framework for epidemic preparedness and response | Epidemic Preparedness Index (EPI) for assessing resilience to epidemic and pandemic outbreaks |
| [10] | i. Health infrastructure<br>q. Urban density<br>f. Disease dynamics<br>r. Economic growth | NA | Literature review and expert elicitation | Identify the most vulnerable countries to infectious disease outbreaks |
| [26] | q. Urban density<br>s. High-density facilities<br>h. Worldwide movement<br>m. Inadequate sanitation | NA | Epidemiology evidence of previous infectious diseases | Identification of specific factors responsible for disease emergence |
| [27] | o. Socioeconomic status<br>c. Age<br>t. Rural or urban living | New Zealand | Epidemiology evidence of previous infectious diseases | Description of vulnerability factors for death in an outbreak of pandemic |
| [2] | u. Public transportation<br>s. Nearby food market<br>o. Overall poverty rate<br>i. Healthcare access<br>v. Public services access | NA | Vulnerable indicators for area classification | Identify geographic areas to be prioritized for preventative action and response efforts |
| [28] | f. Geographic spread<br>k. Infectious period | NA | Demographic, clinical, treatment, and laboratory data | Epidemiological modeling to reduce the disease burden |
| [29] | p. Education levels<br>o. Poor households<br>t. Urbanization<br>q. Population density<br>m. Housing condition<br>i. Health care availability<br>j. Chronic morbidity | India | Epidemiology evidence of COVID-19 | Social vulnerability index for management and mitigation of COVID-19 |

Note: Studies retrieved from the literature search. NA means not applicable.

### 2.2. Statistical Data Analysis

Let $\mathcal{S}$ be a geographical space under investigation (i.e., state, country, or city) defined in terms of a finite set of $N$ smaller spatial units (i.e. countries, census tracts, or zip codes); that is $\mathcal{S} = \{1, 2, \ldots, N\}$. Let $\mathcal{V}$ a set of $M$ vulnerability factors, and $V_k$ the values of the $N$ spatial units in the $k$-th vulnerable factor $V_k = \{v_{k,1}, \ldots, v_{k,N}\}$. The raw data for each factor are normalized across all spatial units over the range 0 (best) to 1 (worst). Different normalization methods exists in the literature [30]. The method chosen in this study was to build an estimation of the Probability Density Function (PDF) of the data, and then transform it via its Cumulative Density Function (CDF), so intervals with higher likelihood of containing data are assigned to higher portion of the normalized interval [0,1]. This is called probability integral transform [31]. We estimate the PDF $f_{V_k}$ at specific spatial unit $x$ using the Kernel Density Estimation (KDE) method.

$$f_{V_k}(x) = \frac{1}{N\lambda} \sum_{i=1}^{N} \mathcal{K}_\lambda(x, v_{k,i}), \tag{1}$$

where $\mathcal{K}$ is the kernel (a non-negative function) and $\lambda$ is the smoothing parameter called the bandwith.

Then, to normalize the raw data at spatial unit $x$ over the range 0 (best) to 1 (worst) in the $k$-th vulnerable factor, the probability integral transform is applied.

$$x' = F_{V_k}(x), \tag{2}$$

where $F_{V_k}$ is the CDF of the $k$-th vulnerable factor.

### 2.3. Cluster Analysis

To identify spatial units with similar levels of vulnerability, areas with similar vulnerability profiles are clustered. Here, the cluster analysis uses the information contained in their vulnerability profile ( expressed from their CDF for the $M$ vulnerability factors) to form spatial groups that were relatively homogeneous in vulnerability, that is, synthesize the spatial units into $k$ partitions.

UVA allows the decision-maker to select the number of cluster partitions in which the spatial units will be grouped (i.e., the selection made according to the number of vulnerability levels desired). Each sub-set of solutions $\mathcal{C} = \{C_1, \dots, C_L\}$ obtained by a cluster algorithm (i.e., k-means) contains a number $N_j$ of spatial units of similar characteristics. In this way, the decision tool makes it possible to obtain a suitable number of $k$ relevant possible vulnerable assessments (i.e., $k = 3$ vulnerability of low, medium, and high; $k = 10$ vulnerability from 1 to 10).

### 2.4. Create Vulnerability Index

To assign a vulnerability level (rank) to each cluster, a Borda's count aggregation method is proposed [32]. The Borda's method takes as input a set of ranks $R = \{R_1, \dots, R_M\}$ (where $R_k$ is an order of the Clusters $\mathcal{C} = \{C_1, \dots, C_L\}$ in the $k$-th vulnerability factor), and produces a single rank by mixing the orders of all the input ranks. The number of points (weight) assigned for each ranking varies depending on which variant of the Borda count is used. For this, let $t_{R_k}^{C_i}$ the position of the Cluster $C_i$ in the rank $R_k$, and $w_{R_k}$ the weight assigned for the rank $R_k$. A new aggregated value of ranking or the $i$-th Cluster is defined as:

$$\mathcal{R}(C_i) = \sum_{k=1}^{M} w_{R_k} \left( |\mathcal{C}| - t_{R_k}^{C_i} \right). \tag{3}$$

To assign a vulnerability level, vulnerability factor ranks $R_k$ were made sorted by the centroid values of $C_i$ for each $M$ vulnerability factors. Next, these $M$ ranks ($R = \{R_1, \dots, R_M\}$) were combined using Borda's count aggregation method to obtain a unique aggregated vulnerability rank.

Finally, the vulnerability rank is associated with a vulnerability index, that is, a higher rank indicates higher vulnerability.

## 3. Vulnerability Index for the COVID-19 in Bogotá, Colombia

### 3.1. Study Area and Data Sources

The UVA presented here was framed in the current COVID-19 pandemic in Bogotá city, the largest and most crowded city in Colombia. Bogotá is a metropolitan city with 7.412.566 inhabitants living in an area of 1775 km (995 km urban and 718 km rural), at an altitude 2640 m, with an annual temperature ranging from 6 to 20 °C, and annual precipitation of over 840 mm. Bogotá has composed of 621 Urban Sectors (Urban Sector is a cartographic division created by the National Administrative Department of Statistics (DANE) [33].). Each Urban sector belongs to one of the 112 Zonal Planning Units (UPZ) [34], see Figure 2.

Information was obtained from the National Department of Statistics (DANE), District Planning Secretary of Bogotá (SDP), and the District Mobility Secretary of Bogotá (SDM). Data comprised public information about demographic, transportation, socio-economic,

and health conditions reported from 2011 to 2020. A summary of the datasets is presented as follows:

- MON_2017 [35,36]: Dataset provided by SDP containing a set monograph which provides a physical, demographic and socioeconomic description of Bogotá and its districts.
- SDM_2017 [37]: Dataset provided by SDM presenting detailed official information of mobility characterization in Bogotá.
- CNPV_2018 [33]: Dataset provided by DANE containing the national census made in 2018 which provides socio-demographic statistics of Colombia.
- DANE_2018 [38]: Dataset provided by DANE containing the results of the Multidimensional Poverty Index which encompasses educational and health quality, work and housing conditions, and access to public services.
- DANE_2020 [39]: Dataset provided by DANE presenting a vulnerability index based on demographic and health conditions relevant for COVID-19 pandemic.

Since the datasets' information are in different spatial units (i.e., Urban sectors, UPZ), we choose the Urban sector for the study. Then, information at the UPZ level is transformed into Urban sectors by spatial transformation (i.e., UPZ values are assigned to each Urban sector contained in this).

### 3.2. Vulnerability Domains

Given the data and the vulnerability factors found in the literature review (see Section 2.1), a set of three relevant domains is proposed: (i) Where and how he/she lives,(ii) Where and how he/she works, and (iii) Where and how he/she moves around (The proposed domains are used for the convenience of the reader and could change depending on the data analysis made in the geographic area. It helps the reader to associate vulnerability factors related. These domains do not influence the process of assigning vulnerability to a spatial unit.). These three domains contain the input for the quantitative analysis. Table 2 shows the domains proposed and the vulnerability factors associated with them.

### 3.2.1. Where and How He/She Lives

Several demographic factors influence the degree of vulnerability of the Urban sector to the pandemic. The literature emphasizes factors such as urban density, age, and the urban living (i.e., socio-spatial segregation). The level of education or literacy, and the quality of the health care system (i.e., included in the poverty index) can also play a helpful role in mitigating the spread and effects of infectious diseases [10]. Further, most data on the COVID-19 pandemic suggest that people with underlying comorbid conditions such as high blood pressure, diabetes, respiratory and cardiovascular disease, and cancer are more vulnerable than people without them.

### 3.2.2. Where and How He/She Works

Urban sectors with high-density facilities (i.e, educational buildings, cultural buildings, sport buildings, food markets, all formal labor) are more vulnerable to the spread of contagious diseases due to space limitations within and between households, growth and mobility, and limited water, sanitation, and hygiene (WASH) infrastructure. Also, most workers in the informal economy (i.e., informal labor) have higher exposure to occupational health and safety vulnerability as they have no appropriate protective equipment, are forced to work daily for their sustenance and must afford all their expenses from cash out-of-pocket due to their limited banking access [40].

### 3.2.3. Where and How He/She Moves Around

Understanding transmissibility, risk of geographic spread, transmission routes, and infection vulnerability factors (i.e., geographic impact) provides the baseline for epidemiological modeling that can inform the planning of response and containment efforts to reduce the burden of disease [28]. Also, there have been claims that the use of public

transport (i.e., public transportation dependency) increase the likelihood of the disease spreading [2].

**Table 2.** Vulnerability domains for the COVID-19 case in Bogotá, Colombia.

| Vulnerability Domains | Vulnerability Factor(s) | Definition | Dataset |
|---|---|---|---|
| Where and how he/she lives | Urban density (q.) | Number of people inhabiting a given urban area | CNPV_2018 |
| | Age (c.) | Number of people aged 15–34 years (SARS-CoV-2 incidence increased [41]) | CNPV_2018 |
| | Comorbidities (j.) | Groups areas according to their demographics and comorbidities | DANE_2020 |
| | Poverty index (p. and i.) | Multiple deficiencies in health, education and standard of living | DANE_2018 |
| | Socio-spatial segregation (t.) | Absence of interaction between individuals of different social groups | [42] * |
| Where and how he/she works | Educational (s.) | Number of educational buildings (i.e., preschool, primary and high-school, research centers, technical training centers, Universities) | MON_2017 |
| | Cultural (s.) | Number of cultural buildings (i.e., theaters, concert halls, libraries, museums, civic centers, community halls) | MON_2017 |
| | Sports (s.) | Number of sports buildings (i.e., stadiums, coliseums, sports clubs, country, racetracks, swimming pools) | MON_2017 |
| | Food markets (s.) | Number of food market buildings (i.e., Central market, market square) | MON_2017 |
| | Formal Labor (s.) | Number of commercial buildings with license | MON_2017 |
| | Informal Labor (s.) | Percentage of informal employed according to its workplace | [43] * |
| Where and how he/she moves around | Public Transportation Dependency (u.) | Number of Trips generated throughout the day (trips longer than 15 min) | SDM_2017 |
| | Transmission routes (f.) | Asymphotmatic number people at the peak of the pandemic | [44] * |
| | Geographic impact (k.) | Number of dead people after 100 simulation days | [44] * |

Note: Letters in the table refer to factors presented in the Section 2.1 (Table 1). * Values calculated in the cited paper.

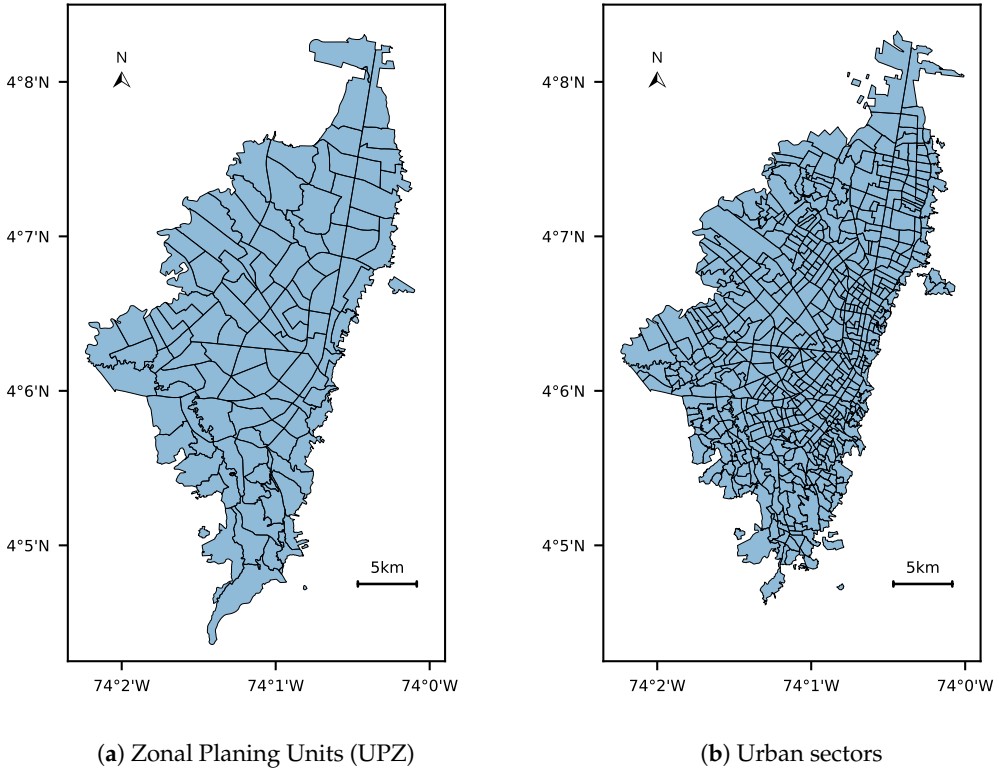

(**a**) Zonal Planing Units (UPZ)　　　　　　　(**b**) Urban sectors

**Figure 2.** Spatial distribution of Bogotá, Colombia using Zonal Planing Units (UPZ) (**a**) and Urban sectors (**b**).

### 3.3. Vulnerability Analysis

To understand the distribution of the vulnerability factors over the Urban sectors, the raw data for each factor is normalized across all Urban sectors over the 0 (less vulnerable) to 1 (most vulnerable) range. Figure 3 shows the normalization for the three domains. The vulnerability value for each factor is associated with the probability integral transform using the KDE method (KDE uses the Gaussian kernel for its estimations and Scott's Rule for the bandwidth selection [45]). The results show the spatial correlation that exists for some vulnerability factors, especially for the Where and how she/he works domain. In contrast to the Where and how she/he lives domain, the spatial correlation is not clear and the vulnerability is distributed across the geography area under study (Bogotá).

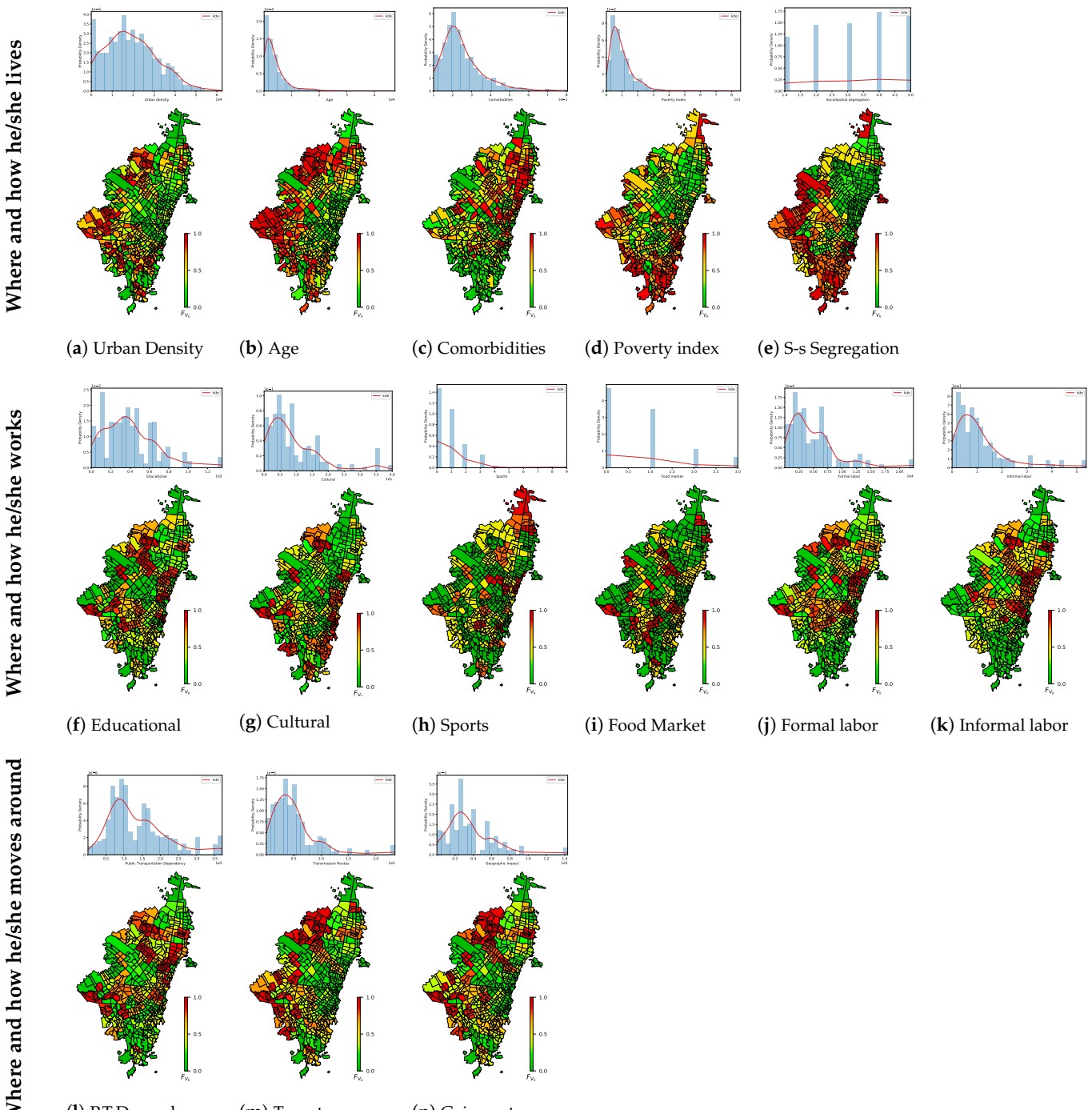

**Figure 3.** Normalization for vulnerability factors using the probability integral transform.

### 3.4. Vulnerability Index

To provide a better vulnerability characterization, the UVA framework generates three different indexes to assess vulnerability in various ways (depending on the cluster partitions). Vulnerability index I has three different clusters that distinguish low, medium, high exposure groups to disease harm. Index II has five different clusters ($k = 5$) to distinguish lowest, low, medium, high, highest vulnerability groups. And, index III has ten clusters ($k = 10$) to represent vulnerability groups on a scale from 1 to 10. Figure 4 shows the three vulnerability indexes.

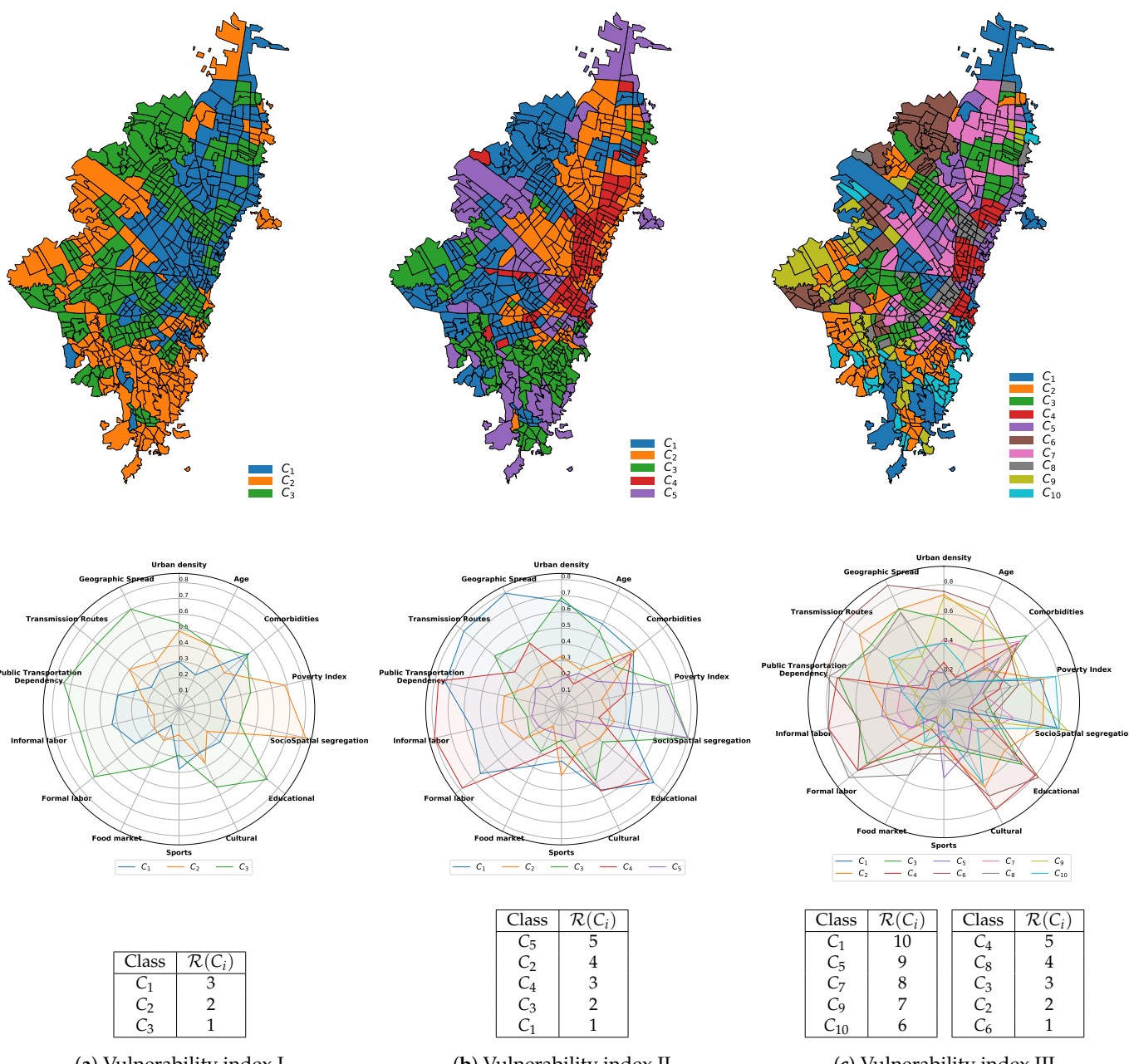

**(a)** Vulnerability index I      **(b)** Vulnerability index II      **(c)** Vulnerability index III

**Figure 4.** Vulnerability index I with $k = 3$ (**a**), index II with $k = 5$ (**b**), and index III with $k = 10$ (**c**). For each Vulnerability index: clusters generated using the k-means method (top), its corresponding centroid values for each vulnerable factor (middle), and the unique rank generated using the Borda's count method (bottom). (The class identifier $1, \ldots, k$ for the clusters of the vulnerability indexes with different $k$ partitions ($k = 3$ left, $k = 5$ medium, $k = 10$ right) does not be the same between models (i.e., the class identifier variate from index to index)).

The clusters' centroids (Figure 4-middle) are used to sort the vulnerability factors in descending order. This sort is interpreted as vulnerability ranking which is used for the analysis. Then, to aggregate the 14 ranks (one for each vulnerable factor in Table 2) in a unique vulnerability ranking the Borda's method is used (Figure 4-bottom). The unique vulnerability ranking is then transformed into a vulnerability index, where a higher rank indicates higher vulnerability. In absence of a rationale for using any weighting, scheme [15,46,47], equal weights were assigned to each vulnerable rank for calculating the overall vulnerability index, according to other studies [29,48].

Figure 5 shows the final three vulnerability index constructed with UVA. In index I, the results show high vulnerable urban sectors in the southwest part of the city. On the other hand, index II shows how some Urban sectors change from medium to low or high-vulnerability, with respect the index I. Further, index III presents an interesting scenario where the spatial correlation between urban sectors is not remarkable getting an unbiased vulnerability index for COVID-19.

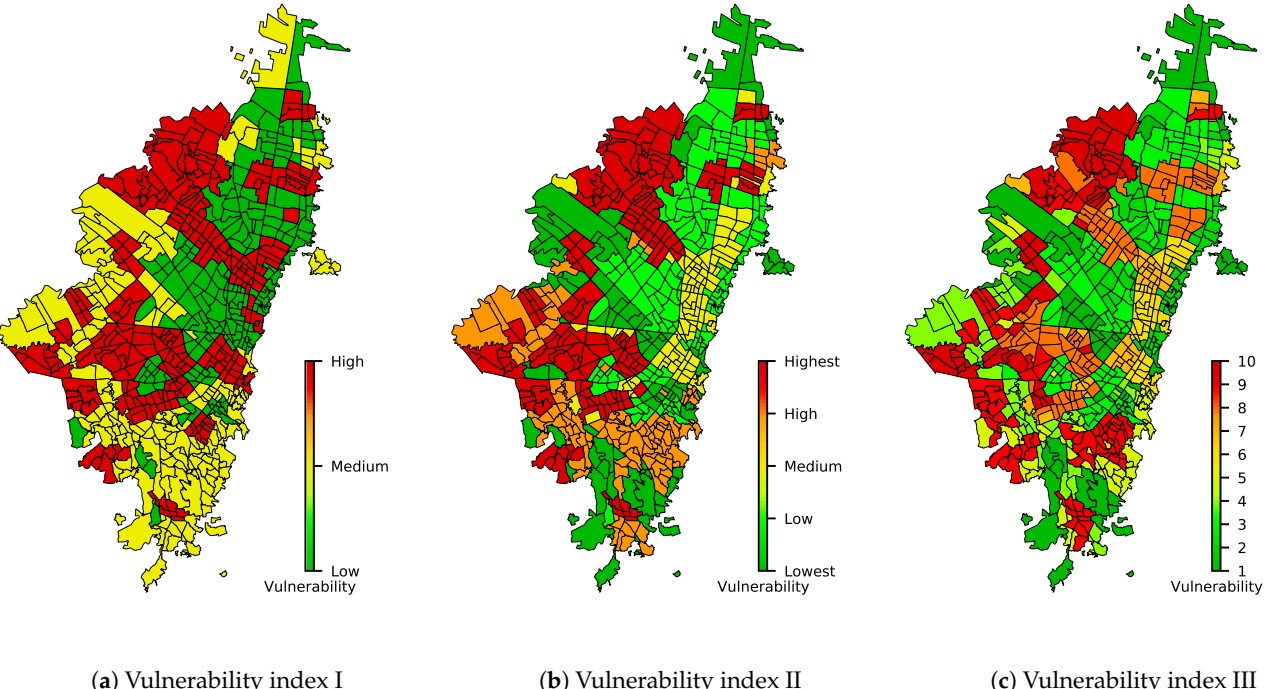

(**a**) Vulnerability index I       (**b**) Vulnerability index II       (**c**) Vulnerability index III

**Figure 5.** Vulnerability indices generated using Urban Vulnerability Assessment (UVA) for the current COVID-19 pandemic in Bogotá, Colombia. Vulnerability index I has 3 levels from low to high (**a**); Vulnerability index II has 5 levels from lowest to highest (**b**); and Vulnerability index III has 10 levels from 1 to 10 (**c**).

Although our intention was not to predict the risk of infection for an Urban sector, we observed some similarities between vulnerability indexes proposed and the current concentration of COVID-19 cases confirmed in Bogotá [49] (Figure 6). The results show how vulnerable areas found with UVA overlap with urban areas with more COVID-19 cases. This indicates that the UVA framework proposed could be used to recommend actions for before, during, and after pandemic that is, to planning and coordination efforts through leadership and coordination across sectors, to assess if the risk of a pandemic could increase in specific geographic areas.

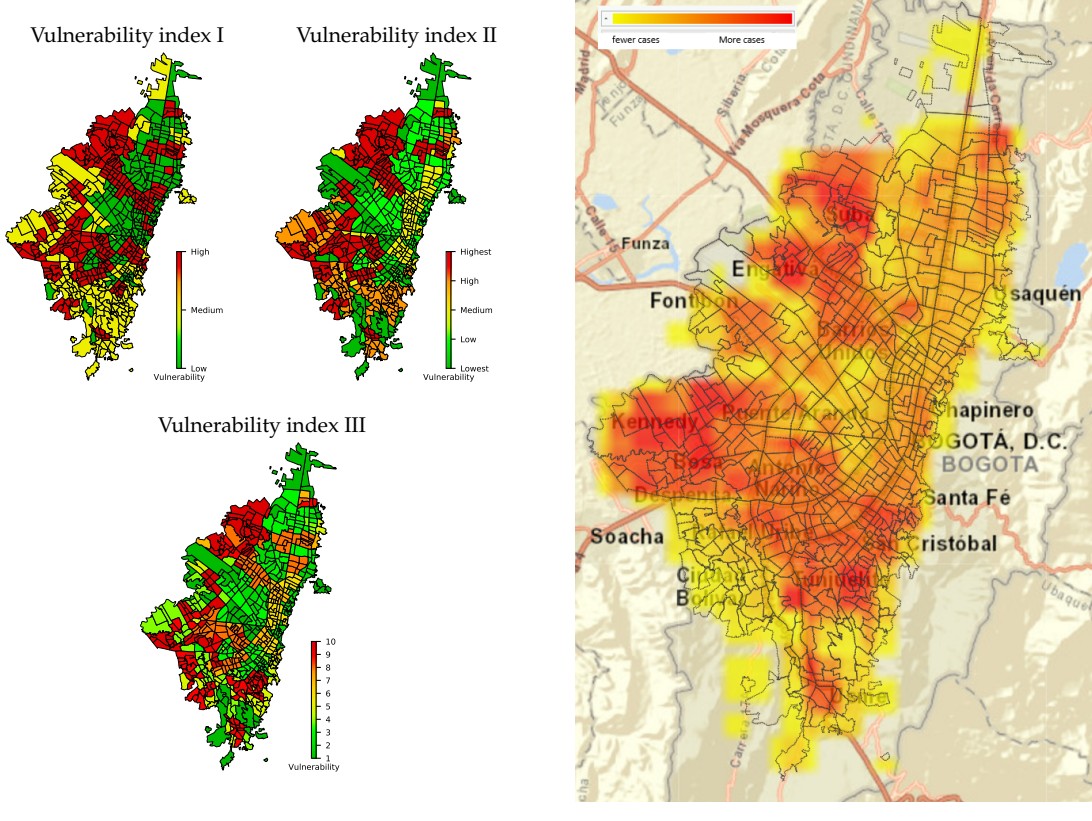

(**a**) Vulnerability indexes proposed in UVA  (**b**) Confirmed COVID-19 cases in Bogotá

**Figure 6.** Comparison between the vulnerability indexes proposed in this study (**a**) and the real COVID-19 cases confirmed in Bogotá (**b**). The colored boxes show the concentrations of the cases in 1000 m on 30 December 2020, in Bogotá, where the red color indicates more confirmed COVID-19 cases and fewer cases are in yellow.)

## 4. Conclusions and Future Work

An Urban Vulnerability Assessment (UVA) for pandemic surveillance is proposed. It was based on a set of 14 vulnerability factors found in the literature. The UVA output defines a composite measure of community-level vulnerability and its spatial distribution, identifying and ranking potentially higher vulnerability areas. The UVA is framed in the current COVID-19 pandemic in Bogotá city, the largest and crowded city in Colombia. UVA creates not only one, but a set of vulnerability indices (i.e., low-high, lowest-highest, and 1–10) to pandemic surveillance.

Although the variables involved in the UVA are structural, the proposed approach is flexible, does not require expert support or knowledge, and allows citizens to be better informed, and policy-makers and international organizations to prioritize resource allocation. Furthermore, the UVA allows to set constraints in short-term city plans (i.e, informing citizens, mandating shelter in place, limit social contact, ban crowds, limit non-essential travel) or long-term scale undertakings (i.e., reduce socio-spatial segregation, decent housing, bio-secure protocols for high-density facilities).

Despite the usefulness of the UVA framework, there are some limitations. Ideally, it would be possible to calculate the index at the neighborhood level. However, several important variables were not available at the neighborhood level. Hence, this analysis is restricted to the urban sector level. On the other hand, the relative importance of the assessment criteria to assign weights to construct the vulnerability index is an issue to be addressed in future research. Also, data used in this study are 1–4 years old and might not have captured vulnerability well in urban sectors in which rapid changes have occurred up to the present day.

Finally, the results suggest a connection between high-vulnerability levels and increased impact and spread of the disease at different geographic levels. Therefore, upon thorough evaluation, UVA could become a relevant tool in the development of policies and programs aimed at reducing disaster risk (DRR) at different city scales (i.e., addressing urban vulnerability at national, regional, and provincial scales), in diverse scenarios of resource scarcity (i.e., short and long-term actions), and for different audiences (i.e., the citizens, policy-makers, international organizations).

**Author Contributions:** Conceptualization, J.P. and R.M.; methodology, J.P. and J.G.; software, J.P.; validation, R.M., J.G., and E.L.; formal analysis, J.P., J.G., and E.L.; investigation, J.P.; resources, J.P.; data curation, J.P. and E.L.; writing—original draft preparation, J.P.; writing—review and editing, J.P., R.M., J.G., and E.L.; visualization, J.P. and E.L.; supervision, R.M., J.G., and E.L.; project administration, J.G., and E.L. All authors have read and agreed to the published version of the manuscript.

**Funding:** This research received no specific grant from any funding agency in the public, commercial, or not-for-profit sectors.

**Conflicts of Interest:** The authors declare no conflict of interest.

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
