# Peer review of "Urban Vulnerability Assessment for Pandemic Surveillance—The COVID-19 Case in Bogotá, Colombia"

_sustainability, doi:10.3390/su13063402_

Round 1
Reviewer 1 Report
Review “Urban Vulnerability Assessment for Pandemic surveillance”
Round 1
- Introduction
- The concept of urban vulnerability assessment (UVA):
- The paper states that “Vulnerability assessment describes the degree to which socioeconomic systems and physical assets in geographic areas are either susceptible or resilient to the impact of disaster (i.e., Pandemic).” Urban vulnerability, however, is an specific form of vulnerability, and its assessment requires of some characteristics to be employed for urban planning. Some insights over this topic can be found in Salas & Yepes (2018) that can be used for providing the reader with an idea of what urban vulnerability is, and why it is important for city planning.
- The authors also indicates that “Besides, it helps to determine what types of preparedness and response activities might help lower the vulnerability classification most quickly.”. However, how urban vulnerability could help in determining which actions should be carried out should be explained.
- Gap:
- The gap is not clearly identified. Instead, it is pointed out that there are some methods dealing with urban vulnerability in urban areas: ”Several models have been proposed to quantify vulnerable geographic areas over the infectious disease domain, i.e., vector-borne diseases [11], Dengue [9], malaria [12,13], and Ebola [10]. More recently, a COVID-19 vulnerability index for urban areas in India was proposed [14].”
- Aiming/contribution:
- “This study proposes a vulnerability Index by aggregating weighted scores of a set of variables related to COVID-19 precaution of social distance and lockdown in geographic areas of four metro cities in India.”
- The novelty of this work should be highlighted in the introduction. It is necessary to identify the gap and explain how the paper contributes bridging it.
- Literature review:
- there are other studies proposing urban vulnerability index related to covid-19 (Mishra et al., 2020; Acharya & Porwal, 2020) that were disregarded by the authors. They should revise the state-of-the-art, and highlighting what is the novelty/contribution of their work to the field of urban vulnerability index related to covid-19.
- Research design and Methodology
The proposed methodology is described in section 2.2., Statistical data analysis, and it consists of several steps. First, data of the city if Bogotá and its districts is collected accordingly to a set of indicators. Then, the districts are grouped into classes (“groups of spatial areas with similar characteristics”) by means of unsupervised cluster analysis (k-means), and each class is ranked for each indicator accordingly to the values of its centroid for each indicator. As a result, the method affords a set of 14 rankings for each of the clusters. Finally, these rankings are aggregated by means of the Borda’s count aggregation method. The whole procedure presents several issues:
- As to the collection of data and indicators to be used in the assessment of UV, the reader expects that the authors will use those stablished in the literature review (section 2.1 of the paper, table 1). However, in section 3.2.1, a different set of indicators (Table 2) is employed. The authors should provide a linkage between the concepts in both tables.
- As to the grouping of geographical areas, there is no explanation about the selection of the number of clusters for the grouping of districts/urban sectors. The k-means method is an unsupervised clustering technique in which the number of clusters is a priori setup by the analyst.
- On the other hand, the proposed methodology builds upon the assumption that in assessing urban vulnerability, all the indicators have the same importance, i.e., all the indicators have the same weighting when being aggregated. However, previous work published in this journal (Salas & Yepes, 2019) has shown that this leads to inaccurate urban vulnerability assessment. In consequence, it is necessary the weighting of urban vulnerability indicators prior to their aggregation, which can be done, for example, by means of multi-criteria analysis (Salas & Yepes, 2019). This issue should be either addressed or the limitation indicated in the paper.
- Case-study/Results
The area analysed is hierarchically structured in two scales, in which the zonal planning units (UPZ) comprises one or more urban sectors.
- Domains of indicators:
- The authors introduce in the case study a series of domains for grouping the indicators that was not introduced neither in the introduction section nor in the methodology. This is relevant since it implies some kind of inner structure in the set of indicators that would affect both to the grouping of urban sections, and to the weighting of rankings for their aggregation. Therefore, it should be included in the methodology and explain how this affects to the assessment process (i.e. if the input data in the cluster process corresponds to all the indicators or only to those of each domain, if the indicators are aggregated by domain to obtain a domain ranking…).
- Vulnerability index
- As in the case of the domains, the authors refer in section 3.4 to a way of assessing urban vulnerability that was not presented in the methodology section. It seems that once they have obtained the rankings, they resort to cluster analysis again for synthesising these rankings into a scale of three, five o ten levels/clusters. The methodology and rationale under this step should be detailed in the methodology section.
- As to the consistency of the results, figure 4 shows that some urban areas (upper-left corner) considered to have medium vulnerability in the 3-level/cluster scale (C2), are classified as having the highest vulnerability in the 5-level/cluster scale (C5), and as having the lowest vulnerability in the 10-level/cluster scale (C1) which results counterintuitive and suggests some kind of inconsistency in the programming/computational process or in the understanding of the results.
- Conclusions
- The conclusions claim that “UVA highlights the vulnerability based on a set of 14 vulnerable factors found in the literature.”. However, as stated in the comments to the case-study, the factors found in the literature review (Table 1) do not correspond to those employed in the UVA (Table 2).
- The results seems to have some inconsistency, as described in comment 4 Vulnerability index, and should therefore be revised.
- The authors indicate that this “UVA could be used to build evidence for planning, modelling, and epidemiological studies to better inform the public, policymakers, and international organizations and funders as to where and how to improve surveillance, response efforts, and delivery of resources, which are crucial factors 208 in containing the COVID-19 Pandemic.”. However, the authors do not provide any example of a potential practical application of the results for decision-making. It would be very helpful in supporting this claim that the authors supply some examples of the usefulness of UVA for decision-making relating covid-19.
- References in the review
- Mishra, S. V., Gayen, A., & Haque, S. M. (2020). COVID-19 and urban vulnerability in India. Habitat International, 103, 102230. http://doi.org/https://doi.org/10.1016/j.habitatint.2020.102230
- Acharya, R., & Porwal, A. (2020). A vulnerability index for the management of and response to the COVID-19 epidemic in India: an ecological study. The Lancet Global Health, 8(9), e1142–e1151. http://doi.org/https://doi.org/10.1016/S2214-109X(20)30300-4
- Salas, J., & Yepes, V. (2018). Urban vulnerability assessment: Advances from the strategic planning outlook. Journal of Cleaner Production, 179, 544–558. http://doi.org/10.1016/j.jclepro.2018.01.088
- Salas, J., & Yepes, V. (2019). VisualUVAM: A Decision Support System Addressing the Curse of Dimensionality for the Multi-Scale Assessment of Urban Vulnerability in Spain. Sustainability, 11(8), 2191. http://doi.org/10.3390/su11082191

Author Response
Reviewer #1:
First, we would like to thank you for providing further constructive suggestions on how to improve the next version. We hope to have addressed these in a satisfactory manner, as explained below, point-by-point. Answers to the queries are in blue and literal changes added to the paper are in red.
1. Introduction
1.a The concept of urban vulnerability assessment (UVA): The paper states that “Vulnerability assessment describes the degree to which socioeconomic systems and physical assets in geographic areas are either susceptible or resilient to the impact of disaster (i.e., Pandemic).” Urban vulnerability, however, is an specific form of vulnerability, and its assessment requires of some characteristics to be employed for urban planning. Some insights over this topic can be found in Salas & Yepes (2018) that can be used for providing the reader with an idea of what urban vulnerability is, and why it is important for city planning. The authors also indicates that “Besides, it helps to determine what types of preparedness and response activities might help lower the vulnerability classification most quickly.”. However, how urban vulnerability could help in determining which actions should be carried out should be explained.
We have included a more detailed explanation of Urban Vulnerability Assessment in the introduction section. We added the following in line 31-34:
In the urban context, the Urban Vulnerability Assessment (UVA) helps to determine what types of preparedness and response activities might help for an optimal Urban Strategic Planning (USP) to assist the decision-making processes undertaken by today’s urban planners [11].
Also, the preventative action and response efforts that should be carried out with the vulnerability assessment are added in line 30-31:
(i.e., planning and coordination, reducing the spread of disease, continuity of health care provision)
1.b Gap: The gap is not clearly identified. Instead, it is pointed out that there are some methods dealing with urban vulnerability in urban areas: ”Several models have been proposed to quantify vulnerable geographic areas over the infectious disease domain, i.e., vector-borne diseases [11], Dengue [9], malaria [12,13], and Ebola [10]. More recently, a COVID-19 vulnerability index for urban areas in India was proposed [14].”
In the section introduction, the gap is explained and we added the following in line 46 - 50:
However, an a-priori knowledge or relative preferences between criteria based judgments for the gathering of preferences for indicators (vulnerable factors) is needed in those models. Different methodologies allows to transform the experts' knowledge into a mathematical language (i.e., Analytic Hierarchy Process), but this methodologies has some limitations such as a-priori knowledge, expert bias, or hierarchical criteria [19, 20]
1.c Aiming/contribution: “This study proposes a vulnerability Index by aggregating weighted scores of a set of variables related to COVID-19 precaution of social distance and lockdown in geographic areas of four metro cities in India.” The novelty of this work should be highlighted in the introduction. It is necessary to identify the gap and explain how the paper contributes bridging it.
Knowing the gap, we extend the idea of the contribution. We added the following in line 53-58:
These factors are used to generate an index that allows us to identify and rank potentially vulnerable urban areas. The rank is built using Borda’s count aggregation method, which does not need experts knowledge nor additional parameters for the construction of the ranking. Then, the vulnerability rank is associated with a vulnerability index, i.e., higher rank indicates higher vulnerability. UVA helps decision-makers to review how the strengthening of surveillance and control measures would mitigate the vulnerability of Pandemic.
Also, we add the value of the study in lines 62-65:
To our knowledge, our study is the first to develop a composite measure of community-level vulnerability concerning the COVID-19 situation in Bogotá. The main value of our study is the Urban sector ranking provided to policy makers to prioritise resource allocation and devise effective mitigation and reconstruction strategies for affected populations in Bogotá.
1.d Literature review: there are other studies proposing urban vulnerability index related to covid-19 (Mishra et al., 2020; Acharya & Porwal, 2020) that were disregarded by the authors. They should revise the state-of-the-art, and highlighting what is the novelty/contribution of their work to the field of urban vulnerability index related to covid-19.
We had already added the Mishra et al., 2020 study in the first version. The Acharya & Porwal, 2020 has been added in Table 1, as a new entry of the literature review.
Also, the contribution is clarified in the reply to comment #2.c.
2. Research design and Methodology
The proposed methodology is described in section 2.2., Statistical data analysis, and it consists of several steps. First, data of the city if Bogotá and its districts is collected accordingly to a set of indicators. Then, the districts are grouped into classes (“groups of spatial areas with similar characteristics”) by means of unsupervised cluster analysis (k-means), and each class is ranked for each indicator accordingly to the values of its centroid for each indicator. As a result, the method affords a set of 14 rankings for each of the clusters. Finally, these rankings are aggregated by means of the Borda’s count aggregation method. The whole procedure presents several issues:
2.a As to the collection of data and indicators to be used in the assessment of UV, the reader expects that the authors will use those stablished in the literature review (section 2.1 of the paper, table 1). However, in section 3.2.1, a different set of indicators (Table 2) is employed. The authors should provide a linkage between the concepts in both tables.
We have included letters for each factor (from a to u) in the Table 1 to group vulnerable factors that are repeated in the literature review. Also, we add in Table 2 ( section 3.2.) the letters of vulnerable factors to link the concepts in both tables.
2.b As to the grouping of geographical areas, there is no explanation about the selection of the number of clusters for the grouping of districts/urban sectors. The k-means method is an unsupervised clustering technique in which the number of clusters is a priori setup by the analyst.
We add a new section called Cluster Analysis where we explain the methodology for grouping geographic areas. We added the following in the lines 114-125:
Cluster Analysis
As a proposal to identify spatial units with possible high levels of vulnerability, a cluster analysis is made to group spatial areas with similar characteristics. Cluster analysis is a multivariate analysis technique that aims to organize information about variables (vulnerable factors) so that relatively homogeneous clusters can be formed i.e., synthesise the spatial units into k partitions. Therefore, each cluster consists of spatial areas with similar behavior from their PDF for the M vulnerable factors}, see Figure 1(c).
UVA allows the decision-maker to select the number of k partitions in which the spatial units will be grouped. Each subset of solution C {= C_1, \dots, C_L} obtained by a cluster algorithm (i.e., k-means) contains a number N_j of spatial units of similar characteristics. In this way, the decision tool makes it possible to get an affordable number of k relevant possible vulnerable assessments (i.e., k=3 vulnerability of low, medium, and high; k=10 vulnerability from 1 to 10).
2.c On the other hand, the proposed methodology builds upon the assumption that in assessing urban vulnerability, all the indicators have the same importance, i.e., all the indicators have the same weighting when being aggregated. However, previous work published in this journal (Salas & Yepes, 2019) has shown that this leads to inaccurate urban vulnerability assessment. In consequence, it is necessary the weighting of urban vulnerability indicators prior to their aggregation, which can be done, for example, by means of multi-criteria analysis (Salas & Yepes, 2019). This issue should be either addressed or the limitation indicated in the paper.
We have clarified this aspect in lines 134-137:
The rank built using Borda’s count aggregation method does not need expert knowledge nor additional parameters for the construction of the ranking (i.e., the weights could be equal for each rank). However, if the rank has to be weighted in some way, the method allows us to assign this weight for each rank.
3. Case-study/Results
The area analysed is hierarchically structured in two scales, in which the zonal planning units (UPZ) comprises one or more urban sectors.
3.a Domains of indicators: The authors introduce in the case study a series of domains for grouping the indicators that was not introduced neither in the introduction section nor in the methodology. This is relevant since it implies some kind of inner structure in the set of indicators that would affect both to the grouping of urban sections, and to the weighting of rankings for their aggregation. Therefore, it should be included in the methodology and explain how this affects to the assessment process (i.e. if the input data in the cluster process corresponds to all the indicators or only to those of each domain, if the indicators are aggregated by domain to obtain a domain ranking…).
We have clarified this aspect in footnote 3:
The proposed domains are used for the convenience of the reader and could change depending on the data analysis made in the geographic area. It helps the reader to associate vulnerability factors. These domains do not influence the process of assigning vulnerability to a spatial unit.
3.b Vulnerability index: As in the case of the domains, the authors refer in section 3.4 to a way of assessing urban vulnerability that was not presented in the methodology section. It seems that once they have obtained the rankings, they resort to cluster analysis again for synthesising these rankings into a scale of three, five o ten levels/clusters. The methodology and rationale under this step should be detailed in the methodology section. As to the consistency of the results, figure 4 shows that some urban areas (upper-left corner) considered to have medium vulnerability in the 3-level/cluster scale (C2), are classified as having the highest vulnerability in the 5-level/cluster scale (C5), and as having the lowest vulnerability in the 10-level/cluster scale (C1) which results counterintuitive and suggests some kind of inconsistency in the programming/computational process or in the understanding of the results.
The section 3.4 was rewritten as follows in lines 210-224:
To provide a better response for vulnerability assessment, UVA generates three different vulnerability indexes to assess vulnerability in different ways (depending on the k partitions). Vulnerability index I has three different clusters ($k=3$) to get a vulnerability index from low to high (i.e., low, medium, high). Vulnerability index II has five different clusters (k=5) to get a vulnerability index from lowest to highest (i.e., lowest, low, medium, high, highest). And, Vulnerability index III has ten clusters (k=10) to get a vulnerability index from 1 to 10. Figure 4 shows the different vulnerability index (i.e., Vulnerability index I with k=3 (3 clusters), Vulnerability index II with k=5 (5 clusters), Vulnerability index III with k=10 (10 clusters)).
After getting the clusters for each vulnerability index (showed in Figure 4 - top), the centroids of the clusters (showed in Figure 4 - midst) are getting and used to sort from higher to lower values the each vulnerability factor. These vulnerability factors sorted (by the centroids) are assumed as vulnerable ranks that would be used for the analysis. Then, to aggregate the 14 ranks (one for each vulnerable factor in Table 2) the Borda's count aggregation method build a unique vulnerability ranking for each cluster (showed in Figure 4 - bottom).
Also, footnote 6 was added to clarify the Clusters identifiers.
The class identifier $1, \dots, k$ for the clusters of the vulnerability indexes with different k partitions (k=3 left, k=5 medium, k=10 right) does not be the same between models (i.e., the class identifier variate from index to index).
4. Conclusions
4.1 The conclusions claim that “UVA highlights the vulnerability based on a set of 14 vulnerable factors found in the literature.”. However, as stated in the comments to the case-study, the factors found in the literature review (Table 1) do not correspond to those employed in the UVA (Table 2).
We have clarified this aspect in the reply to comment #2.a.
4.2 The results seems to have some inconsistency, as described in comment 4 Vulnerability index, and should therefore be revised.
We have clarified this aspect in the reply to comment #3.a.
4.3 The authors indicate that this “UVA could be used to build evidence for planning, modelling, and epidemiological studies to better inform the public, policymakers, and international organizations and funders as to where and how to improve surveillance, response efforts, and delivery of resources, which are crucial factors 208 in containing the COVID-19 Pandemic.”. However, the authors do not provide any example of a potential practical application of the results for decision-making. It would be very helpful in supporting this claim that the authors supply some examples of the usefulness of UVA for decision-making relating covid-19.
We add another figure (Figure 6.) that shows an example of UVA in real-world settings. Also, we added the following in the lines 232-238:
Finally, the Figure 6. compares the vulnerability indexes proposed in this study and the real COVID-19 cases confirmed in Bogotá. The results show how vulnerable areas found with UVA matches with urban areas with more COVID'19 cases. This indicates that the UVA framework proposed could be used to recommend actions for before, during and after pandemic i.e., to planning and coordination efforts through leadership and coordination across sectors, to assess if the risk of a pandemic could increase in specific geographic areas.
Reviewer 2 Report
The paper proposes an interesting study in line with the topics of this journal. The authors developed a new methodology to evaluate the urban vulnerability by considering the possible vulnerable factors related to the spread of Covid-19 pandemics to identifying the more vulnerable urban areas. The methodology (Urban Vulnerability Assessment) was developed considering the previous studies outcomes on the vulnerability in the urban areas related to the past pandemics experience and factors found in the current pandemic. Also, it was applied to the urban area of Bogotá, Colombia.
Following some suggestions to improve the quality of the paper before the publication:
- Explain the importance to use a vulnerability factor analysis in this type of study;
- Support the methodological chooses with adequate bibliographic citations. In particular, explain the used criteria to select this type of statistical analysis (Kernel Density Estimation and Borda's method for the cluster) in the methodology;
- Figure 3, 4 and 5 must resize. The maps and the graphs are not clear and legible;
- Add a new paragraph to explain in details the outcomes of the application to the Bogotá urban area;
- Improve paragraph 4 with more considerations on the methodology's usefulness for ordinary urban planning practices and managing emergencies.
Author Response
Reviewer #2:
First, we would like to thank you for providing further constructive suggestions on how to improve the next version. We hope to have addressed these in a satisfactory manner, as explained below, point-by-point. Answers to the queries are in blue and literal changes added to the paper are in red.
1. Explain the importance to use a vulnerability factor analysis in this type of study;
We have included a more detailed explanation of Urban Vulnerability Assessment in the introduction section. We added the following in line 31-34:
In the urban context, the Urban Vulnerability Assessment (UVA) helps to determine what types of preparedness and response activities might help for an optimal Urban Strategic Planning (USP) to assist the decision-making processes undertaken by today’s urban planners [11].
Also, the preventative action and response efforts that should be carried out with the vulnerability assessment are added in line 30-31:
(i.e., planning and coordination, reducing the spread of disease, continuity of health care provision)
2. Support the methodological chooses with adequate bibliographic citations. In particular, explain the used criteria to select this type of statistical analysis (Kernel Density Estimation and Borda's method for the cluster) in the methodology;
We have clarified the KDE aspect in lines 103-108:
Different normalization methods exists in the literature [29]. However, depending on the data properties, some normalization operations are inappropriate (i.e., anomalies, orthogonality, linear dependency). One solution is to build an estimation of the Probability Density Function (PDF) of the data, and then transform it via its Cumulative Density Function (CDF), so intervals with higher likelihood of containing data are assigned to higher portion of the normalized interval [0,1]. This is call probability integral transform [30].
We add a new section called Cluster Analysis where we explain the methodology for grouping geographic areas. We added the following in the lines 114-125:
Cluster Analysis
As a proposal to identify spatial units with possible high levels of vulnerability, a cluster analysis is made to group spatial areas with similar characteristics. Cluster analysis is a multivariate analysis technique that aims to organize information about variables (vulnerable factors) so that relatively homogeneous clusters can be formed i.e., synthesise the spatial units into k partitions. Therefore, each cluster consists of spatial areas with similar behavior from their PDF for the M vulnerable factors}, see Figure 1(c).
UVA allows the decision-maker to select the number of k partitions in which the spatial units will be grouped. Each subset of solution C {= C_1, \dots, C_L} obtained by a cluster algorithm (i.e., k-means) contains a number N_j of spatial units of similar characteristics. In this way, the decision tool makes it possible to get an affordable number of k relevant possible vulnerable assessments (i.e., k=3 vulnerability of low, medium, and high; k=10 vulnerability from 1 to 10).
We have clarified the borda aspect in lines 134-137:
The rank built using Borda’s count aggregation method does not need expert knowledge nor additional parameters for the construction of the ranking (i.e., the weights could be equal for each rank). However, if the rank has to be weighted in some way, the method allows us to assign this weight for each rank.
3. Figure 3, 4 and 5 must resize. The maps and the graphs are not clear and legible;
Fixed, thank you.
4. Add a new paragraph to explain in details the outcomes of the application to the Bogotá urban area;
We add the value of the study in lines 62-65:
To our knowledge, our study is the first to develop a composite measure of community-level vulnerability concerning the COVID-19 situation in Bogotá. The main value of our study is the Urban sector ranking provided to policy makers to prioritise resource allocation and devise effective mitigation and reconstruction strategies for affected populations in Bogotá.
Also, we add another figure (Figure 6.) that shows an example of UVA in real-world settings. Also, we added the following in the lines 232-238:
Finally, the Figure 6. compares the vulnerability indexes proposed in this study and the real COVID-19 cases confirmed in Bogotá. The results show how vulnerable areas found with UVA matches with urban areas with more COVID'19 cases. This indicates that the UVA framework proposed could be used to recommend actions for before, during and after pandemic i.e., to planning and coordination efforts through leadership and coordination across sectors, to assess if the risk of a pandemic could increase in specific geographic areas.
5. Improve paragraph 4 with more considerations on the methodology's usefulness for ordinary urban planning practices and managing emergencies.
We improve the conclusion as follows in lines 247-262:
Surveillance is of primary importance to monitor the burden of disease and will give both local authorities and the global community a chance for a quick response to public health threats.
Our work has demonstrated how high-vulnerable level contexts contribute to increase the impact and spread of the disease at different geographic levels. This approach also enables the design of comprehensive plans, implemented at city scale, for addressing urban vulnerability at national, regional, and provincial scales.
Round 2
Round 2
The authors have successfully addressed many of the issues pointed out in the previous review. However, there are still some problems arising from the methodology they employ in the aggregation of ranks. They argue that the Bourda’s method for aggregating ranking does not require of any a priori knowledge-assumption, which is truth, but does not mean that the problem of aggregating ranks of criteria with different importance is solved. On the contrary, by applying the Bourda’s method to assess urban vulnerability, the authors are obliterating the fact that in urban vulnerability assessment, there are some criteria that are more important in representing/quantifying urban vulnerability (Salas & Yepes, 2019). As they themselves acknowledge, “The rank built using Borda’s count aggregation method does not need expert knowledge nor additional parameters for the construction of the ranking (i.e., the weights could be equal for each rank). However, if the rank has to be weighted in some way, the method allows us to assign this weight for each rank.”. In their work, the authors have implicitly assumed (as an a-priori knowledge) that the weights are equal for each rank, which is a very hard to sustain assumption that should be explicitly stated and pointed out as a limitation. To make their methodology fully representative of urban vulnerability, they must provide the Borda’s count aggregation with the weighting they are referring to.
Therefore, in order to make their manuscript acceptable, I suggest the authors that they indicate this (the assumption that all criteria have the same importance in assessing urban vulnerability or, in other words, the lack of an evaluation of the relative importance between the assessment criteria) as a limitation to be addressed through future research.
Introduction
- The concept of urban vulnerability assessment (UVA):
- Gap:
Review 1: The gap is not clearly identified. Instead, it is pointed out that there are some methods dealing with urban vulnerability in urban areas: ”Several models have been proposed to quantify vulnerable geographic areas over the infectious disease domain, i.e., vector-borne diseases [11], Dengue [9], malaria [12,13], and Ebola [10]. More recently, a COVID-19 vulnerability index for urban areas in India was proposed [14].”
Authors: “However, an a-priori knowledge or relative preferences between criteria based judgments for the gathering of preferences for indicators (vulnerable factors) is needed in those models. Different methodologies allows to transform the experts' knowledge into a mathematical language (i.e., Analytic Hierarchy Process), but this methodologies has some limitations such as a-priori knowledge, expert bias, or hierarchical criteria [19, 20] “
Review 2: the proposed methodology does not overcome the problem of requiring some a-priori knowledge for the weighting of criteria in the aggregation since it is assuming that all of them have the same weighting. Therefore, they are not achieving the objective of bridging the identified gap. A swift in the gap is required to make it consistent with the contribution (e.g., “To our knowledge, our study is the first to develop a composite measure of community-level vulnerability concerning the COVID-19 situation in Bogotá”).
Aiming/contribution:
Review 1: The novelty of this work should be highlighted in the introduction. It is necessary to identify the gap and explain how the paper contributes bridging it.
Authors: “These factors are used to generate an index that allows us to identify and rank potentially vulnerable urban areas. The rank is built using Borda’s count aggregation method, which does not need experts knowledge nor additional parameters for the construction of the ranking. Then, the vulnerability rank is associated with a vulnerability index, i.e., higher rank indicates higher vulnerability. UVA helps decision-makers to review how the strengthening of surveillance and control measures would mitigate the vulnerability of Pandemic.”
Review 2: The approach employed by the authors assume that all criteria have the same importance/weight, a hypothesis that it is not only supported but contrary to current research on urban vulnerability assessment methods. It is not that the borda’s method does not need of expert judgment for the construction of the ranking, but this is because Borda’s method does not account, in fact, for different importance between criteria, which makes this method inappropriate for the elicitation of a composite-aggregated ranking of urban vulnerability (unless the authors assume that all criteria have the same contribution to urban vulnerability, which is untenable). I therefore suggest that the authors point out that the rank is built using Borda’s count aggregation method under the assumption that all criteria have the same importance, and then, in the conclusions section, they indicate the need of evaluating the relative importance between criteria (Salas & Yepes, 2019) as an issue to be addressed in future research.
Case-study/Results
- Domains of indicators:
- Vulnerability index
Review 1: As in the case of the domains, the authors refer in section 3.4 to a way of assessing urban vulnerability that was not presented in the methodology section. It seems that once they have obtained the rankings, they resort to cluster analysis again for synthesising these rankings into a scale of three, five o ten levels/clusters. The methodology and rationale under this step should be detailed in the methodology section.
Authors: “To provide a better response for vulnerability assessment, UVA generates three different vulnerability indexes to assess vulnerability in different ways (depending on the k partitions). Vulnerability index I has three different clusters ($k=3$) to get a vulnerability index from low to high (i.e., low, medium, high). Vulnerability index II has five different clusters (k=5) to get a vulnerability index from lowest to highest (i.e., lowest, low, medium, high, highest). And, Vulnerability index III has ten clusters (k=10) to get a vulnerability index from 1 to 10. Figure 4 shows the different vulnerability index (i.e., Vulnerability index I with k=3 (3 clusters), Vulnerability index II with k=5 (5 clusters), Vulnerability index III with k=10 (10 clusters)).
After getting the clusters for each vulnerability index (showed in Figure 4 - top), the centroids of the clusters (showed in Figure 4 - midst) are getting and used to sort from higher to lower values the each vulnerability factor. These vulnerability factors sorted (by the centroids) are assumed as vulnerable ranks that would be used for the analysis. Then, to aggregate the 14 ranks (one for each vulnerable factor in Table 2) the Borda's count aggregation method build a unique vulnerability ranking for each cluster (showed in Figure 4 - bottom).”
Review 2: It would be of help that, in order to make meaningful the selectin of the number of clusters, the authors indicate in their description of cluster analysis (response 2.b) that this selection is made accordingly to the number of levels desired by the analyst to grade UVA.
Conclusions
References in the review
- Salas, J., & Yepes, V. (2019). VisualUVAM: A Decision Support System Addressing the Curse of Dimensionality for the Multi-Scale Assessment of Urban Vulnerability in Spain. Sustainability, 11(8), 2191. http://doi.org/10.3390/su11082191

Author Response
Reviewer #1:
First, we would like to thank you for providing further constructive suggestions on how to improve the next version. We provide a few answers and clarifications for each outstanding comment below. Answers to the queries are in blue and literal changes added to the paper are in red.
0. The authors have successfully addressed many of the issues pointed out in the previous review. However, there are still some problems arising from the methodology they employ in the aggregation of ranks. They argue that the Bourda’s method for aggregating ranking does not require of any a priori knowledge-assumption, which is truth, but does not mean that the problem of aggregating ranks of criteria with different importance is solved. On the contrary, by applying the Bourda’s method to assess urban vulnerability, the authors are obliterating the fact that in urban vulnerability assessment, there are some criteria that are more important in representing/quantifying urban vulnerability (Salas & Yepes, 2019). As they themselves acknowledge, “The rank built using Borda’s count aggregation method does not need expert knowledge nor additional parameters for the construction of the ranking (i.e., the weights could be equal for each rank). However, if the rank has to be weighted in some way, the method allows us to assign this weight for each rank.”. In their work, the authors have implicitly assumed (as an a-priori knowledge) that the weights are equal for each rank, which is a very hard to sustain assumption that should be explicitly stated and pointed out as a limitation. To make their methodology fully representative of urban vulnerability, they must provide the Borda’s count aggregation with the weighting they are referring to.
Therefore, in order to make their manuscript acceptable, I suggest the authors that they indicate this (the assumption that all criteria have the same importance in assessing urban vulnerability or, in other words, the lack of an evaluation of the relative importance between the assessment criteria) as a limitation to be addressed through future research.
Indeed, this aspect wasn’t clear in the paper and we thank the reviewer for pointing it out. The following part has been removed from the section 2.4 Create Vulnerability index to generalize the Borda’s method.
The rank built using Borda’s count aggregation method does not need experts knowledge nor additional parameters for the construction of the ranking (i.e., the weights could be equal for each rank). However, if the rank has to be weighted in some way, the method allows assigning this weight for each rank.
Also, we add a paragraph with the limitations in lines 250-256:
Despite the usefulness of the UVA framework, there are some limitations. Ideally, it would be possible to calculate the index at the neighborhood level. However, several important variables were not available at the neighborhood level. Hence, this analysis is restricted to the urban sector level. Furthermore, the relative importance of the assessment criteria to assign weights to construct the vulnerability index is an issue to be addressed in future research. Finally, data used in this study are 1–4 years old and might not have captured vulnerability well in urban sectors in which rapid changes have occurred up to the present day.
1. Introduction
The concept of urban vulnerability assessment (UVA):
1.a Gap:
Review 2: The proposed methodology does not overcome the problem of requiring some a-priori knowledge for the weighting of criteria in the aggregation since it is assuming that all of them have the same weighting. Therefore, they are not achieving the objective of bridging the identified gap. A swift in the gap is required to make it consistent with the contribution (e.g., “To our knowledge, our study is the first to develop a composite measure of community-level vulnerability concerning the COVID-19 situation in Bogotá”).
To clarify the use of the same weights for the rank, we add the following in lines 223-225
In absence of a rationale for using any weighting scheme [18,44,45], equal weights were assigned to each vulnerable rank for calculating the overall vulnerability index, following the studies presented in [28,46].
Also, the word “a-priori knowledge” was removed from the paper to avoid confusion.
1.b Aiming/contribution:
Review 2: The approach employed by the authors assume that all criteria have the same importance/weight, a hypothesis that it is not only supported but contrary to current research on urban vulnerability assessment methods. It is not that the borda’s method does not need of expert judgment for the construction of the ranking, but this is because Borda’s method does not account, in fact, for different importance between criteria, which makes this method inappropriate for the elicitation of a composite-aggregated ranking of urban vulnerability (unless the authors assume that all criteria have the same contribution to urban vulnerability, which is untenable). I therefore suggest that the authors point out that the rank is built using Borda’s count aggregation method under the assumption that all criteria have the same importance, and then, in the conclusions section, they indicate the need of evaluating the relative importance between criteria (Salas & Yepes, 2019) as an issue to be addressed in future research.
We have clarified this aspect in the reply in comment #0 and #1.a.
2. Case-study/Results
2.1 Vulnerability index:
Review 2: It would be of help that, in order to make meaningful the selection of the number of clusters, the authors indicate in their description of cluster analysis (response 2.b) that this selection is made accordingly to the number of levels desired by the analyst to grade UVA.
We have clarified this aspect added the following in section 2.3 Cluster Analysis in line 123:
(i.e., the selection made accordingly to the number of vulnerability levels desired)
Reviewer 2 Report
The revisions made by the authors can be considered sufficient for the publication of the article
Author Response
Thanks for your comments and valuable help.
Round 3
Reviewer 1 Report
The authors have properly addressed the issues raised in the last review. I found, however, that the conclusions contain an statement that is not fully supported by the results: I refer to the sentence "Our work has demonstrated how high-vulnerable level contexts contribute to increasing the impact and spread of the disease at different geographic levels.". At the end of section 3.4 Vulnerability Index, the authors indicate that "Although our intention was not to predict the risk of infection for an Urban sector, we observed similarities between vulnerability indexes proposed and the current concentration of COVID-19 cases confirmed in Bogotá, see Figure 6." However, this "simmilarity" is not evidence enough for the reader to consider demonstrated a relationship between high-vulnerability levels and increased impact and spread of the disease, which would requiere of further work via statistical analysis. I therefore recommend the authors that they point out the relevance of their finding by indicating in the conclussions that the results suggest a strong connection between high-vulnerability levels and increased impact and spread of the disease.
Author Response
Reviewer #1:
First, we would like to thank you for providing further constructive suggestions on how to improve the next version. We hope to have addressed these in a satisfactory manner, as explained below, point-by-point. Answers to the queries are in blue and literal changes added to the paper are in red.
1. The authors have properly addressed the issues raised in the last review. I found, however, that the conclusions contain an statement that is not fully supported by the results: I refer to the sentence "Our work has demonstrated how high-vulnerable level contexts contribute to increasing the impact and spread of the disease at different geographic levels.". At the end of section 3.4 Vulnerability Index, the authors indicate that "Although our intention was not to predict the risk of infection for an Urban sector, we observed similarities between vulnerability indexes proposed and the current concentration of COVID-19 cases confirmed in Bogotá, see Figure 6." However, this "simmilarity" is not evidence enough for the reader to consider demonstrated a relationship between high-vulnerability levels and increased impact and spread of the disease, which would requiere of further work via statistical analysis. I therefore recommend the authors that they point out the relevance of their finding by indicating in the conclussions that the results suggest a strong connection between high-vulnerability levels and increased impact and spread of the disease.
Indeed, this aspect wasn’t clear in the paper and we thank the reviewer for pointing it out. The word "demonstrated" was removed for the paper and the final paragraph of the conclusions was updated in lines 257-258.
Our work suggests a strong connection between high-vulnerability levels and increased impact and spread of the disease at different geographic levels. This would enable the design of comprehensive plans, implemented at the city scale, for addressing urban vulnerability at national, regional, and provincial scales.